# The Myokine Irisin Promotes Osteogenic Differentiation of Dental Bud-Derived MSCs

**DOI:** 10.3390/biology10040295

**Published:** 2021-04-03

**Authors:** Francesca Posa, Graziana Colaianni, Michele Di Cosola, Manuela Dicarlo, Francesco Gaccione, Silvia Colucci, Maria Grano, Giorgio Mori

**Affiliations:** 1Department of Clinical and Experimental Medicine, University of Foggia, Viale Pinto 1, 71122 Foggia, Italy; francesca.posa@unifg.it (F.P.); dr.dicosola@gmail.com (M.D.C.); 2Department of Emergency and Organ Transplantation, University of Bari, 70124 Bari, Italy; graziana.colaianni@uniba.it (G.C.); manuela.dicarlo25@gmail.com (M.D.); fra.gaccione@gmail.com (F.G.); maria.grano@uniba.it (M.G.); 3Department of Basic Medical Sciences, Neuroscience and Sense Organs, Section of Human Anatomy and Histology, University of Bari, 70124 Bari, Italy; silviaconcetta.colucci@uniba.it

**Keywords:** bone, irisin, osteocalcin, mesenchymal stem cells (MSCs), precision medicine, dental stem cells, osteoblastogenesis, translational research

## Abstract

**Simple Summary:**

Irisin is a recently discovered protein, mainly produced in the muscle tissue, whose action is proving effective in many other tissues. The crosstalk between muscle and bone has been long since demonstrated, and physical activity has shown to have an impressive positive effect in both tissues. Irisin production increases with exercising and drops with sedentariness and aging, indicating that the molecule is involved in sarcopenia and in bone mass reduction. Although skeleton is target of irisin, its mechanism of action on bone cells has not yet been completely elucidated. The aim of this work is to analyze the effect of irisin on osteoblast differentiation; to this purpose, we used a stem cell model reproducing the osteoblastogenesis and the bone-forming processes. We performed an in vitro study exploring the main osteoblast markers in the presence of irisin. We found that irisin has an impressive effect on the most peculiar osteoblast feature: the bone mineral matrix secretion process. Moreover, irisin demonstrated an inductive effect on osteoblast osteocalcin production. Both results suggest a stimulating effect of irisin in bone formation. The association we observed between irisin addition and osteoblast osteocalcin production should be further investigated.

**Abstract:**

The myokine irisin, well known for its anabolic effect on bone tissue, has been demonstrated to positively act on osteoblastic differentiation processes in vitro. Mesenchymal stem cells (MSCs) have captured great attention in precision medicine and translational research for several decades due to their differentiation capacity, potent immunomodulatory properties, and their ability to be easily cultured and manipulated. Dental bud stem cells (DBSCs) are MSCs, isolated from dental tissues, that can effectively undergo osteoblastic differentiation. In this study, we analyzed, for the first time, the effects of irisin on DBSC osteogenic differentiation in vitro. Our results indicated that DBSCs were responsive to irisin, showed an enhanced expression of osteocalcin (OCN), a late marker of osteoblast differentiation, and displayed a greater mineral matrix deposition. These findings lead to deepening the mechanism of action of this promising molecule, as part of osteoblastogenesis process. Considering the in vivo studies of the effects of irisin on skeleton, irisin could improve bone tissue metabolism in MSC regenerative procedures.

## 1. Introduction

Mesenchymal stem cells (MSCs) are extensively used as a therapeutic resource in modern medicine; they are self-renewable and can differentiate into all cell lineages that form mesenchymal and connective tissues. MSCs can be applied in the treatment of numerous diseases, either alone or integrated with scaffolds; their efficacy has been reported in the therapy of wounds [1], bone and cartilage dehiscence [2], graft-versus-host disease [3], as well as cardiovascular [4] and neural disorders [5].

Stem cell populations present in the bone marrow are the most studied MSCs; other possible alternatives reside in many different parts of the organism, such as adipose tissue, as well as brain, liver, and dental tissues [6].

Dental tissues such as dental pulp, apical papilla, and periodontal ligament are recognized as a good source of MSCs; in particular, dental pulp is a connective tissue with an abundance of stem cells in the perivascular niche, which is similar to the embryonal mesenchyme [7,8].

It has been demonstrated that dental pulp stem cells (DPSCs) express most of the osteogenic markers, thus they can be considered very similar to osteoblast precursors [9,10].

The dental bud (DB) is the undifferentiated stage of the tooth, and as such it is rich in undifferentiated stem cells. The dental papilla, which we find in the mesenchymal part of the DB, contains a very large number of MSCs, more than the dental pulp; these cells express the characteristic mesenchymal stem markers and demonstrate an impressive osteoblastic phenotype [11].

The osteogenic features of dental bud stem cells (DBSCs) can be positively influenced by several molecules, such as vitamins, natural compounds, or drugs [12,13,14,15,16].

The recently identified myokine irisin has been demonstrated to induce periodontal ligament cell (PDLC) growth, migration, and osteogenic differentiation [17] and also to promote the odontogenic differentiation of DPSCs [18].

Irisin has been previously described as a circulating myokine secreted from skeletal muscle in response to exercise. Later, the skeleton was identified as the principal target of irisin that is involved in both osteoblast proliferation [19] and differentiation [20,21].

Thus, irisin plays an important role in bone remodeling; moreover, in vivo studies on mice models have shown that low doses of irisin, administered intermittently, improve mineral density and some important mechanical bone features [22,23,24].

DBSCs, when properly cultured, undergo osteogenic differentiation, expressing the typical markers and forming mineralized matrix; therefore, they can be considered an excellent osteoblastogenesis model. In this study, we investigate the role of irisin on our model of osteoblastogenesis.

## 2. Materials and Methods

### 2.1. Dental Bud Stem Cells (DBSCs) Cultures

Dental buds (DBs) were isolated from the third molar of ten pediatric patients (8–12 years) undergoing dental extractions for orthodontic reasons, after receiving their parents’ informed consent.

The inner part of the DB, the dental papilla, was cut into small pieces and enzymatically digested, using a digesting medium containing 3 mg/mL Type I collagenase and 4 mg/mL dispase, until a suspension consisting of single cells was obtained [25]. DBSCs were seeded and expanded in vitro as previously described [7,9,11,26,27].

In the experiments aimed to examine the effect of irisin on DBCSs, cells were seeded with a density of 15 × 10^3^/cm^2^ in Dulbecco’s modified Eagle’s medium (D-MEM) (Thermo Fisher Scientific, Waltham, MA, USA) supplemented with 5% heat-inactivated fetal bovine serum (FBS), 1% penicillin/streptomycin (Thermo Fisher Scientific, Waltham, MA, USA), in 6-well plates. After 2 days of culture, DBSCs were starved for at least 3 h and subsequently stimulated with 100 ng/mL of irisin for the different time points.

Then, we induced DBSCs osteogenic differentiation in vitro: cells were seeded with a density of 5 × 10^3^/cm^2^ in 24-well plates (for staining) and 6-well plates (for Western blot), cultured in the osteogenic medium consisting of D-MEM supplemented with 2% FBS, 10^−8^ M dexamethasone, and 50 µg/mL ascorbic acid. For the evaluation of mineralized matrix nodule formation, 10 mM β-glycerophosphate (Sigma Aldrich, Milan, Italy) was also added to the osteogenic medium. Cells cultured in these conditions were treated or not (Ctr) with irisin, added to the culture medium at every change (every 3 days).

### 2.2. Irisin Treatment

Irisin (Adipogen, Liestal, Switzerland), was reconstituted at 100 µg/mL in Milli-Q water and stored at −20 °C before use. We treated one part of the cells with a concentration of 100 ng/mL irisin (treatment group), added to the medium. The part of the cells that was not treated with irisin served as the control group (Ctr).

### 2.3. Western Blot

Revelation of pERK at protein levels was performed using SDS-PAGE gel electrophoresis and Western blot analysis. After DBSC starvation in serum-free D-MEM for at least 3 h, irisin was added or not (Ctr) to the medium. Undifferentiated DBSCs, after 2 days of culture, were stimulated with irisin for 0, 5, 7, 10, or 20 min; differentiated DBSCs, after 10 days of osteogenic differentiation, were treated with irisin for 5 or 10 min. At the different time points, cells were lysed as previously described [11,14]. Once the total protein concentration was determined, by using a protein assay (Bio-Rad Laboratories, Hercules, CA, USA), the same amounts of protein for each sample were separated via SDS-PAGE and transferred to nitrocellulose membranes (Invitrogen, Carlsbad, CA, USA). Blots were then probed with primary and secondary antibodies. Anti-pERK and anti-total ERK antibodies were purchased from Santa Cruz Biotechnology (Dallas, TX, USA). The visualization of the immune complexes was carried out using the Odyssey Infrared Imaging System of LI-COR (LI-COR Biotechnology, Lincoln, NE, USA).

### 2.4. Real-Time PCR

DBSCs were cultured in 6-well plates and the total RNA was extracted utilizing spin columns (RNeasy, Qiagen, Hilden, Germany) after 3 or 8 h of stimulation with irisin. Then, 2 μg of RNA was reverse transcribed (RT) by using SuperScript First-Strand Synthesis System kit (BioRad iScript Reverse Transcription Supermix). Afterwards, 20 ng of the synthesized cDNA were used for the quantitative PCR. Real-time PCR analysis was carried out using the SsoFast EvaGreen Supermix (Bio-Rad Laboratories, Hercules, CA, USA) on a CFX96 Real-Time System (Bio-Rad Laboratories, Hercules, CA, USA) following the manufacturer’s protocol. Gene expression was calculated using the mean cycle threshold value (Ct) from triplicate samples, normalizing the results to the average of β-actin, GAPDH, and β_2_ microglobulin (B2M) levels for each reaction.

The following primer pairs were used for the Real Time-PCR (RT-PCR) amplification (Table 1).

### 2.5. Alkaline Phosphatase (ALP)

The expression of ALP, the osteoblastic differentiation marker, was determined with the Leukocyte Alkaline Phosphatase kit (Sigma Aldrich, Milan, Italy) after 7 days of DBSC osteogenic differentiation.

Briefly, once the culture medium was removed, cells were fixed at room temperature for 5 min, following the manufacturer’s protocol. Then the cells were rinsed with deionized water and stained with ALP solution (FRV-Alkaline Solution, Naphthol AS-BI Alkaline Solution, and Sodium Nitrite Solution) for 15 min in the dark. At the end of the incubation time, the wells were washed again and left to dry in the air, then the cells were examined on a microscope. ALP-positive cells show a purple staining.

### 2.6. Alizarin Red Staining (ARS)

In order to test the possible involvement of irisin in inducing the production of calcium-rich deposits in vitro, ARS was carried out on DBSCs differentiated in osteogenic medium for 21 days. Once the culture medium was removed cells were rinsed with PBS and fixed in 10% formalin at room temperature for 10 min. Then, cells were washed with deionized water and stained with a 1% ARS solution (Alizarin Red powder was purchased from Sigma Aldrich, Milan, Italy) for 10 min at room temperature. The ARS solution was discarded at the end of the incubation time, and cells were rinsed twice with deionized water and then air dried.

### 2.7. Statistical Analysis

Data are representative of three/five independent experiments. Boxplots with median and interquartile ranges show data in figures with three values per group. Box-and-whisker plots with median and interquartile ranges, from max to min, show data in figures with four or five values per group. All data points are shown. Statistical calculations were performed using GraphPad Prism version 8.0.2 for MacOS (GraphPad Software, La Jolla, CA, USA, www.graphpad.com). The results were considered statistically significant for *p* < 0.05. ImageJ software (Research Services Branch, Image Analysis Software Version 1.53a, NIH, Bethesda, MD, USA) was used to process images.

## 3. Results

### 3.1. Extracellular Signal-Regulated Kinase (ERK) Phosphorylation in DBSCs Treated with Irisin

In order to clarify whether DBSCs were responsive to the treatment with the myokine or not, we performed experiments on the signaling pathways, focusing on ERK phosphorylation.

DBSCs were seeded in D-MEM supplemented with 5% FBS, cultured for 2 days, starved for at least 3 h, and then stimulated with irisin for the different time points. Whole cell lysates were collected and assessed by Western blot. As shown in Figure 1, pERK expression was transiently upregulated after irisin stimulation. ERK phosphorylation was significantly upregulated after 5 and 7 min of irisin administration, compared to the time zero (T0), and then its level decreased, becoming comparable to the T0. These results demonstrated that DBSCs are responsive to irisin treatment by increasing pERK. 

In parallel, DBSCs were cultured in the osteogenic medium and, after 10 days of differentiation, were treated with irisin for 3 or 5 min. Interestingly, the cells appeared to be still responsive to irisin treatment and even more sensitive than the undifferentiated ones, showing a peak in the phosphorylation of ERK at 3 min (Figure 2), which already decreased after 5 min stimulation.

### 3.2. Irisin Stimulation Increases Osteocalcin (OCN) Expression in DBSCs

Once the responsiveness of our cell model to irisin was confirmed, in order to find out if this myokine could affect the DBSC acquisition of osteoblastic features, we assessed the mRNA expression level of the main osteoblastic markers performing RT-PCR. DBSCs cultured in the appropriate conditions were stimulated or not (Ctr) with irisin for 3 or 8 h and, following this treatment, we identified the most exciting effect of irisin in OCN expression. Figure 3 shows that OCN mRNA levels, although comparable between Ctr and treatment at 3 h, greatly increased in the cells treated with irisin after 8 h of stimulation. This enhancement in OCN expression also turned out to be statistically significant. Interestingly, for all the other osteoblastic markers analyzed (i.e., RANK-L, osteoprotegerin (OPG), collagen I, RUNX-2, and bone sialoprotein (BSP)), we did not reveal any significant change in mRNA expression level at 3 or 8 h of irisin treatment. Nevertheless, it should be noted that the expression of some of these markers appeared to follow a growing trend in the treatment compared to the control (Figure 3, RANK-L, RUNX-2, and BSP graphs), particularly after 8 h of stimulation.

### 3.3. ALP Positivity and Calcium-Rich Deposit Formation in DBSCs

In order to examine the ability of irisin to induce DBSC differentiation in osteoblasts, cells were cultured in osteogenic medium and stimulated with 100 ng/mL of irisin, which was added to the medium at every renewal (every 3 days). DBSC cultures were stopped at 7 and 21 days of continuous irisin treatment in differentiating conditions.

After 7 days of continuous stimulation, we evaluated DBSC alkaline phosphatase (ALP) expression by using the ALP staining. We found that there were only slight differences between Ctr and irisin treatment in the expression of this typical osteoblastic marker (Figure 4a), which were not significant (Figure 4b).

Furthermore, long-term cultures proved the ability of DBSCs, when suitably differentiated in an osteogenic sense, to form calcium-rich deposits in vitro. This process, which characterizes the final step of matrix production by osteoblasts, was assessed with ARS quantification after 21 days of osteogenic differentiation. Interestingly, DBSC deposition of mineral matrix nodules was greater when cells were treated with irisin, compared to the control (Figure 5a,b). These observations were confirmed by the quantitative analysis of the percentage area covered by the red staining (Figure 5c), which was significantly higher in the treatment with irisin when compared to the Ctr.

## 4. Discussion

Both myokine irisin and MSCs have elicited worldwide interest for their extensive potential to treat a large array of translational clinical indications, allowing ample discussions and debates regarding their possible applications in the forthcoming era of precision medicine.

To the best knowledge of the authors, this is the first time that the influence of irisin, a powerful myokine for bone tissue physiology, has been investigated on MSCs from dental bud.

We previously reported that DBSCs constitute a perfect model of osteoblastogenesis [11], thus the mesenchymal part of the bud is a great source of MSCs that can acquire osteogenic features. In this study, we analyzed the effects of irisin on our cell model, examining the expression of the main markers of osteoblast differentiation and focusing on DBSC activity.

All the anabolic effects of irisin on bone have being explored only in recent years, and its functional receptor is still a critical point of study. It has been recently shown that irisin acts on osteocytes by binding to an αv integrin receptor and promoting osteocytes survival [28], but there is still a number of unresolved questions regarding the mechanism. Among the primary signaling events that occur when an integrin receptor is activated by the binding of its ligand, there is the stimulation of the ERK cascade [29]. The phosphorylation of ERK has been already demonstrated as a key pathway in osteoblastic proliferation and differentiation induced by irisin [19].

Firstly, we demonstrated a cellular response, with the phosphorylation of ERK following irisin administration, indicating that DBSCs are responsive to irisin.

Irisin is known to directly stimulate osteoblast proliferation, differentiation, and mineralization in vitro, by increasing ERK phosphorylation [19,20]. More recently, irisin has also been demonstrated to promote matrix formation in PDLCs [17] and hDPSCs [18].

Our results are in line with the data from previous literature and show, for the first time, that DBSCs respond to irisin stimulation; moreover, it was observed that the cells maintained their responsiveness to the myokine even when they were cultured in osteogenic conditions, and induced to differentiate to osteoblasts. Afterwards, we also observed that the mineral matrix deposition was significantly increased in DBSCs cultured in osteogenic condition and continuously treated with irisin. The production of calcium-rich deposits is the most peculiar osteoblast feature, and MSCs or DBSCs, opportunely differentiated, can effectively secrete a mineralized matrix in vitro [11,30].

On the contrary, irisin did not affect ALP activity in DBSCs. Although this observation is not in accordance with the demonstrated effect of irisin on hDPSC ALP activity [18], it should be underlined that Son et al. made use of a much higher concentration of the molecule (20 μM compared to 8.3 nM) normally used in such in vitro studies, and therefore is of little relevance in physiological terms [31]. 

Successively, collagen I, RUNX-2, BSP, OPG, and RANK-L expression were studied. Although irisin appeared to induce the expression of some of these osteogenic markers, no significant variation was found.

Remarkably, a significant myokine effect was detected on OCN expression level in DBSCs. OCN is a non-collagenic glycoprotein, present in bone matrix and circulating in blood, which is involved in osteoblast mineralization process and, in the undercarboxylated form, also in many other organism pathways [32].

Osteoblasts secrete OCN in bone matrix, where it binds to the calcium ions in hydroxyapatite, which is its mineral component. The role of OCN in the mineralization process has been extensively discussed in bone research, with conflicting results indicating both an inhibitory and stimulating effect [33,34,35,36].

Moreover, a previous study showed an existing positive correlation between serum levels of irisin and OCN in a population of healthy children [37]. Interestingly, in a population of older adult subjects, OCN mRNA levels measured in bone biopsies were positively associated with the mRNA expression for the irisin precursor, FNDC5, in skeletal muscle biopsies [38]. These results, obtained in humans, agreed with in vitro data showing that long-term treatment with recombinant irisin raised OCN mRNA expression in primary mouse and rat osteoblasts [19,22].

Recent studies showed that OCN has a crucial role in bone matrix formation since it binds to hydroxyapatite and forms complex with osteopontin, which is an essential adhesion molecule for both osteoblasts and osteoclasts [39,40,41]. OCN hence acts as a bridge between calcium crystals and structural proteins, exerting its function at the organic–inorganic interface of bone matrix, and might serve to inhibit crack, stretching and therefore dissipating energy in collagen fibers [42,43]. Consistently with this issue, fracture resistance is reduced in OCN null mice [44].

## 5. Conclusions

Thus, in conclusion, we can assert that the results of our study highlight a significant effect of irisin in increasing OCN expression in DBSCs. This outcome, together with the rise in matrix mineralization, indicates a major involvement of irisin in the osteogenic differentiation process of DBSCs. In addition, considering that our model resembles the osteoblastogenesis process, we can speculate that irisin administration could contribute to bone matrix deposition by the stimulation of OCN expression. Further in vivo studies will need to be performed in order to confirm this hypothesis.

## Figures and Tables

**Figure 1 biology-10-00295-f001:**
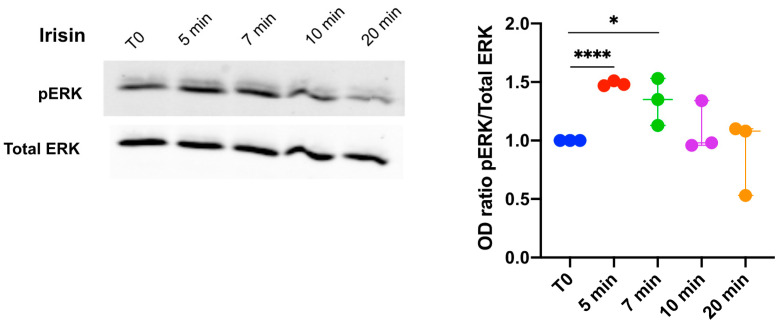
Effects of irisin on extracellular signal-regulated kinase (ERK) phosphorylation (pERK) in DBSCs cultures. Dental bud stem cells (DBSCs) were stimulated with irisin for the indicated time points. Phosphorylated and total ERK expression levels in cell lysates were examined by Western blot and measured with corresponding densitometry band quantification. Densitometric analysis is expressed as the relative optical density (OD) of the bands normalized to total ERK. The graph shows that the treatment significantly, but transiently, increased the protein expression level of pERK after 5 and 7 min of stimulation. The graph represents data as a boxplot with median and interquartile ranges of *n* = 3 independent experiments. * *p* < 0.05, **** *p* < 0.0001. Statistics: unpaired Student’s *t*-test.

**Figure 2 biology-10-00295-f002:**
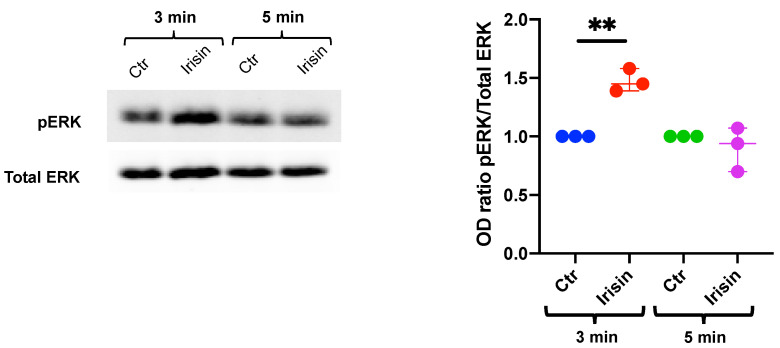
Effects of irisin on pERK in differentiated DBSCs. DBSCs, differentiated in osteogenic medium for 10 days, were stimulated with irisin for 3 or 5 min. pERK and total ERK expression levels in cell lysates were examined by Western blot and measured with corresponding densitometric quantification. Densitometric analysis is expressed as the relative optical density (OD) of the bands normalized to total ERK. The graph shows that the treatment significantly increased the protein expression level of pERK after 3 min of stimulation. The graph represents data as a boxplot with median and interquartile ranges of *n* = 3 independent experiments. ** *p* < 0.01. Statistics: unpaired Student’s *t*-test.

**Figure 3 biology-10-00295-f003:**
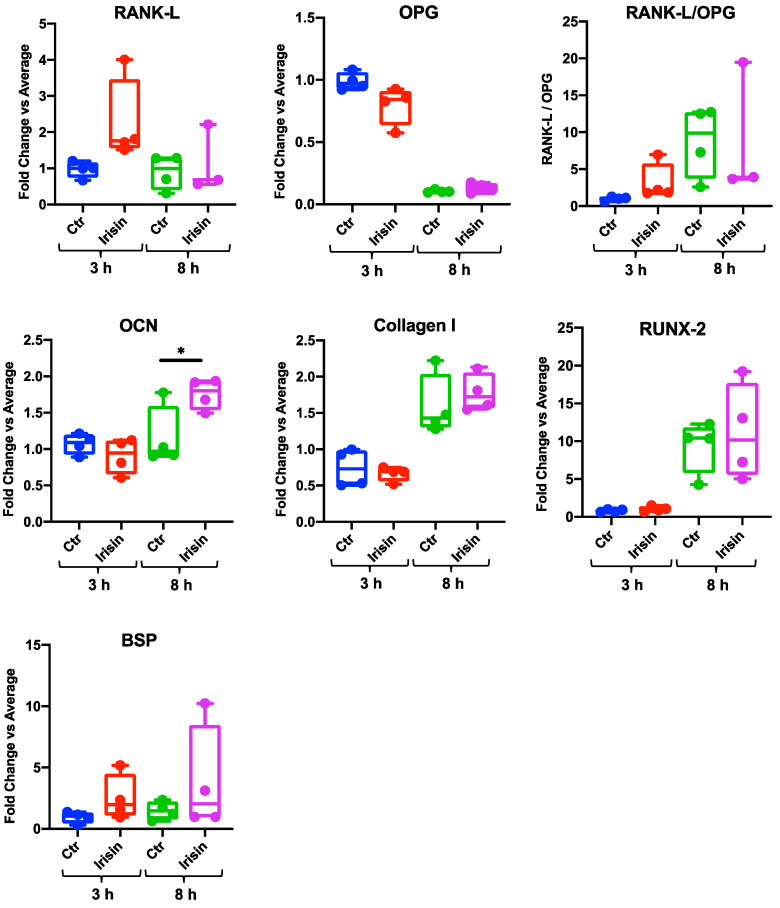
Effects of irisin on the expression of osteoblast markers at mRNA level. mRNA expression levels of osteoblast marker genes (RANK-L, osteoprotegerin (OPG), osteocalcin (OCN), collagen I, RUNX-2, and bone sialoprotein (BSP)) were assayed (qPCR) after 3 and 8 h of stimulation with irisin. The graph shows that the treatment significantly enhanced the expression of the osteoblast marker OCN. Expression was normalized to the average of β-actin, β_2_ microglobulin (B2M), and GAPDH levels for each reaction. Graphs represent data as box-and-whisker plots with median and interquartile ranges, from max to min, of four independent experiments performed in triplicate. * *p* < 0.05 compared to Ctr. Student’s *t*-test was used for single comparisons.

**Figure 4 biology-10-00295-f004:**
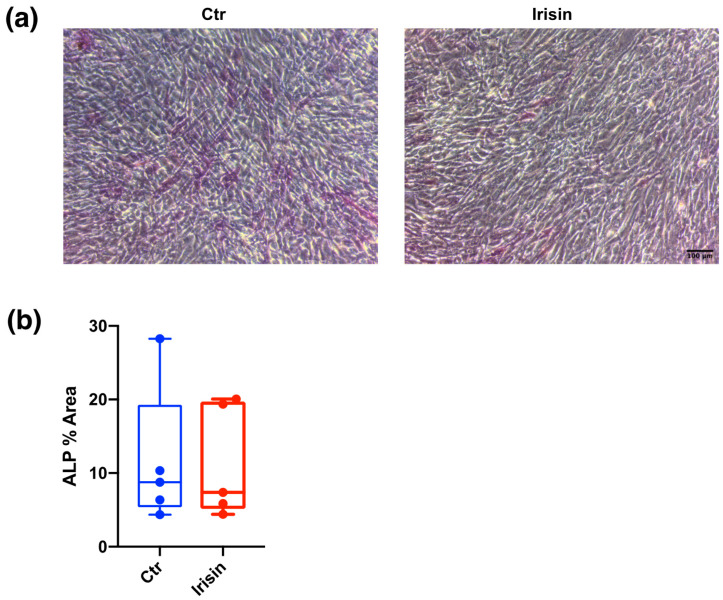
Alkaline phosphatase (ALP) expression in DBSCs. (**a**) ALP expression assayed by ALP cytochemical assay (purple staining) in DBSCs cultured in osteogenic conditions and treated with irisin or not (Ctr) for 7 days. Representative phase contrast pictures were chosen for the figure. Scale bar: 100 μm. (**b**) The graph shows the quantification of ALP % area presented as box-and-whisker plots with median and interquartile ranges, from max to min, and is representative for five independent experiments performed in quadruplicates.

**Figure 5 biology-10-00295-f005:**
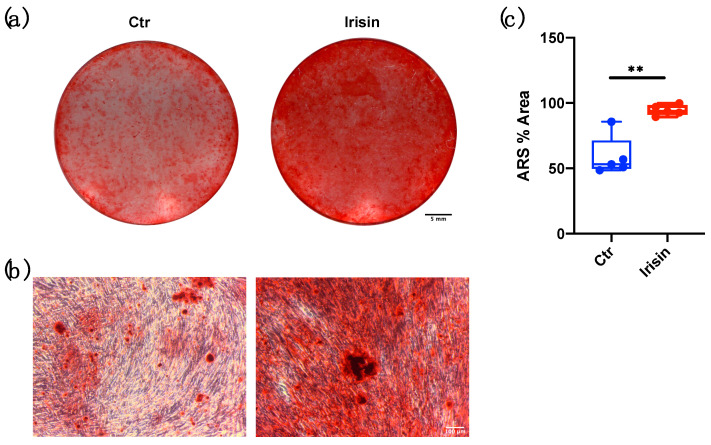
Mineralized nodules formation following irisin stimulation and alizarin red staining (ARS) quantification. (**a**) Mineral matrix deposition, assessed by ARS (red staining), in DBSCs cultured in osteogenic conditions and treated with irisin or not (Ctr) for 21 days. The figure shows the wells of a representative experiment. Scale bar: 5 mm. (**b**) Phase contrast pictures displaying the formation of calcified nodules. Scale bar: 100 μm. (**c**) The graph shows the quantification of ARS % area presented as box-and-whisker plots with median and interquartile ranges, from max to min, and is representative for five independent experiments performed in quadruplicates. ** *p* < 0.01. Statistics: unpaired Student’s *t*-test.

**Table 1 biology-10-00295-t001:** List of primer sequences used for RT-PCR.

Gene	Sense (5′–3′)	Antisense (5′–3′)
RANK-L	ACAGCACATCAGAGCAGAG	AGGACAGACTCACTTTATGGG
OPG	CAAAGGCAGGCGATACTTCC	ATGGAGATGTCCAGAAACACGA
OCN	CTCACACTCCTCGCCCTATTG	GCTTGGACACAAAGGCTGCAC
Coll I (COL1A1)	TGAAGGGACACAGAGGTTTCAG	GTAGCACCATCATTTCCACGA
RUNX-2	CGCCTCACAAACAACCACAG	ACTGCTTGCAGCCTTAAATGAC
BSP	CAATCTGTGCCACTCACTGC	TTTGGTGATTGCTTCCTCTGG
β-Actin (ACTB)	AATCGTGCGTGACATTAAG	GAAGGAAGGCTGGAAGAG
β2 Microglobulin (B2M)	ATGAGTATGCCTGCCGTGTGA	GGCATCTTCAAACCTCCATG
GAPDH	GGAGTCAACGGATTTGGT	GTGATGGGATTTCCATTGAT

## Data Availability

The raw/processed data required to reproduce these findings are available on request from the corresponding author. The data are not publicly available due to technical or ethical limitations.

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
