# Peer review of "The Myokine Irisin Promotes Osteogenic Differentiation of Dental Bud-Derived MSCs"

_biology, 2021, doi:10.3390/biology10040295_

Round 1

Reviewer 1 Report

Summary

This study aimed to investigate the effects of the myokine Irisin on  Dental Bud Stem Cells (DBSCs).  First, the authors demonstrated   a cellular response, with the phosphorylation of ERK following Irisin administration, indicating that DBSCs are responsive to Irisin. Then they studied the effects of Irisin on the expression of osteoblast markers at mRNA level., and discovered that Irisin promoted osteocalcin mRNA expression, while it did not significantly affect those of Collagen I, RUNX-2, BSP, OPG, RANK-L. They  evaluated DBSC alkaline phosphatase (ALP) expression by using the ALP staining and showed that DBSC ALP expression was not significantly affected by Irisin.  Finally they demonstrated that DBSC deposition of mineral matrix nodules was greater in cells cultured with Irisin, compared to the control. 

Major comments

As for Figure 1, pERK expression was not so significantly upregulated after 7 minutes of Irisin stimulation. The authors should present other data which demonstrates that pERK expression was significantly upregulated after  7 minutes of Irisin stimulation. Otherwise, they should rewrite text.

In this study, while Irisin promoted osteocalcin mRNA expression, it did not significantly affect other osteoblast differentiation markers (Collagen I, RUNX-2, BSP, OPG, RANK-L). The manuscript would be improved if the authors could present data which shows the reason(s).  At least they must discuss about the mechanism in the text. 

The manuscript would be improved if the authors could present data which shows the effects of Irisin on osteocalcin promotor activities, by using luciferase assay system.

Minor comments

The authors should show the effect of Irisin on mRNA expression level of ALP. 

Author Response

Reviewer #1

Summary

This study aimed to investigate the effects of the myokine Irisin on  Dental Bud Stem Cells (DBSCs).  First, the authors demonstrated   a cellular response, with the phosphorylation of ERK following Irisin administration, indicating that DBSCs are responsive to Irisin. Then they studied the effects of Irisin on the expression of osteoblast markers at mRNA level., and discovered that Irisin promoted osteocalcin mRNA expression, while it did not significantly affect those of Collagen I, RUNX-2, BSP, OPG, RANK-L. They  evaluated DBSC alkaline phosphatase (ALP) expression by using the ALP staining and showed that DBSC ALP expression was not significantly affected by Irisin.  Finally they demonstrated that DBSC deposition of mineral matrix nodules was greater in cells cultured with Irisin, compared to the control.

We would like to thank the reviewer for the thorough review of our manuscript and the constructive criticisms.

Major comments

  1. As for Figure 1, pERK expression was not so significantly upregulated after 7 minutes of Irisin stimulation. The authors should present other data which demonstrates that pERK expression was significantly upregulated after 7 minutes of Irisin stimulation. Otherwise, they should rewrite text.

Response 1: We thank the reviewer for the remark and we would also like to explain that figure 1 shown in the manuscript is a representative image referred to 1 of 3 independent experiments. As the reviewer can see in the WB original images (herein attached), the effect of Irisin at 7 min is evident in the other two WBs which we preferred not to show in the manuscript. The densitometric analysis also confirmed a statistical significance (p < 0.05) in the upregulation of pERK at 7 min. Probably the image we chose in the manuscript does not allow to clearly see the increase of pERK expression at 7 min because, at that time point, also the loading control (TotalERK) is reduced, when compared to those observed at all the other time points. If the reviewer deems it appropriate, we could also replace the image shown in Figure 1 with the one related to the 3rd WB, which we had discarded because less clean and, above all, because it also contained the unstarved sample which we removed in subsequent experiments.

The images are in the uploaded file

  1. In this study, while Irisin promoted osteocalcin mRNA expression, it did not significantly affect other osteoblast differentiation markers (Collagen I, RUNX-2, BSP, OPG, RANK-L). The manuscript would be improved if the authors could present data which shows the reason(s).  At least they must discuss about the mechanism in the text.
  2. The manuscript would be improved if the authors could present data which shows the effects of Irisin on osteocalcin promotor activities, by using luciferase assay system.

Response 2 and 3: We thank the reviewer for bringing up the discussion regarding the modulation of the osteoblastic markers also related to comment 3.

The direct effect of Irisin (short time points, i.e. 3 and 8 hrs) on typical osteoblastic markers, such as RUNX-2, BSP and RANK-L has never been proven before. On the other hand, Irisin influence on Coll I (Colaianni G. et al. Irisin enhances osteoblast differentiation in vitro. International journal of endocrinology 2014, doi:https://doi.org/10.1155/2014/902186) and OPG (Colucci S. et al. Irisin prevents microgravity-induced impairment of osteoblast differentiation in vitro during the space flight CRS-14 mission. FASEB J. 2020, doi: 10.1096/fj.202000216R) expression was already investigated, but always using cell cultures treated in presence of osteogenic differentiation factors (L-Ascorbic Acid and ?-Glycerophosphate). These observations demonstrate that the inductive effect of Irisin on osteoblastic markers is supported by osteogenic factors. Therefore, the intent of the present study is to investigate the effect of the myokine on MSCs, without any osteogenic factor which could support the myokine action, and we found no important modulation of the main osteogenic markers, but intriguingly a significant effect on Osteocalcin expression.

Furthermore, at present there is no evidence whether there is a direct effect on the Osteocalcin gene promoter, but it is demonstrated that PGC‐1α, a well-known regulator of FNDC5 transcription (Norheim, F. et al. (2014), The effects of acute and chronic exercise on PGC‐1α, irisin and browning of subcutaneous adipose tissue in humans. FEBS J, 281: 739-749. https://doi.org/10.1111/febs.12619), cooperates with NURR1 for binding to the OCN promoter. It is certainly an interesting aspect to dissect in future studies and we intent to work on this.

Minor comments

  1. The authors should show the effect of Irisin on mRNA expression level of ALP.

Response 4: In all our studies on DBSCs, we have always used ALP staining as a reliable indicator of osteoblastic differentiation. Also this time we analyzed the expression of ALP using a histochemical assay and, since we found no differences between control and treatment, we did not deepen the analysis at the RNA level. Surely this aspect could be investigated in future studies.

Reviewer 2 Report

  1. Why the Irisin didn't upregulate the ALP expression in DBSCs when compared to the control group?
  2. The authors claimed that "Further in vivo studies will need to be performed in order to confirm this hypothesis". The reviewer wants to know how will the in vivo studies be designed.
  3. For qPCR assay, why chose 3 and 8 hrs as the time points? Please cite reference. 

Author Response

Reviewer #2

  1. Why the Irisin didn't upregulate the ALP expression in DBSCs when compared to the control group?

Response 1: Probably because ALP is not the only molecule involved in matrix mineralization, after all many osteo-inductive events are not accompanied by ALP increase.

It might be that the effect of Irisin is manifest in DBSCs mineralization process rather than in their differentiation, since ALP is the first event of osteoblast differentiation. This could be the reason why, although there was no substantial difference between control and treatment in ALP expression, an induction of OCN expression and of mineralization degree was clear.

  1. The authors claimed that "Further in vivo studies will need to be performed in order to confirm this hypothesis". The reviewer wants to know how will the in vivo studies be designed.

Response 2: Irisin will be administered IP in mouse models and MSCs from Bone Marrow and dental tissues will be harvested and studied for osteogenic features.

  1. For qPCR assay, why chose 3 and 8 hrs as the time points? Please cite reference.

Response 3: The aim of the present study was to identify the early response of Irisin on our cell model of MSCs. We have already used these time points for the evaluation of the osteoblastic marker expression in response to Irisin treatment and they allowed us to observe clear modulations (Colaianni G, et al. The myokine irisin increases cortical bone mass. Proc Natl Acad Sci U S A. 2015; doi: 10.1073/pnas.1516622112). These scientific evidences led us to use the same methodology in parallel for carrying out the experiments here presented, in order to analyze the direct effect of Irisin.

Reviewer 3 Report

The myokine Irisin promotes osteogenic differentiation of dental derived MSCs: The authors investigated the effects of Irisin on MSCs from Dental Bud Stem Cells (DBSCs), showing its ability to promote osteoblastic differentiation. In particular, they observed enhanced expression of Osteocalcin (OCN), a late marker of osteoblast differentiation, thus suggesting an improvement of osteoblast mineralizing process in presence of Irisin. The authors propose Irisin as promising molecule participating in osteoblastogenesis process, and conclude that it could be used to improve bone tissue regeneration starting from dental stem cells.

Major comments:

Too ambitious a conclusion (“The authors propose Irisin as promising molecule participating in osteoblastogenesis process, and conclude that it could be used to improve bone tissue regeneration starting from dental stem cells”), in relation to the simple tests performed in the present work.

Bone mineral matrix secretion and production of osteocalcin do not suggest an inductive effect of Irisin in osteoblastogenesis (dependent on the activation of osteoblast precursors to committed osteoblasts) but in improvement of osteoblast differentiation. The authors do not clarify whether there is a positive effect of irisin on increased recruitment of osteoblast precursors, rather than increased/accelerated committed-osteoblast differentiation. The undoubted demonstration is related to the “involvement of Irisin in the osteogenic differentiation process of DBSCs”.

Proposed Title: The myokine Irisin promotes osteogenic differentiation of dental bud-derived MSCs.

Self-citations: 15 out of the 44 (34%, more than one third of the total). Such a high number of self-citations is fine for a review on a topic historically carried out by the authors, but I don't think it is appropriate for an original article that is quite simple.

Moreover, some self-citations are not relevant to the demonstration of irisin effect on osteoblastic differentiation: [23], [19].

  • [19]: … induce the “browning” response stimulating the trans-differentiation of white adipocytes
  • [23]: … effect of Irisin in increasing the polar moment of inertia of bones of mice subjected to Irisin administration.

Minor comments:

Introduction and M&M are well described

Results: What is reported in lines 195-199 and 202-204 is more suitable for the discussion paragraph rather than the results paragraph.

Discussion and Conclusions:

Recent studies showed that OCN could influence the bone matrix microarchitecture: Please, explain better. Thus, OCN tightly binds to hydroxyapatite and forms complex structures with collagen and the glycoprotein Osteopontin; OCN could be the link between the organic component of matrix and the mineral fraction: The organic–inorganic interface in bone is not “microarchitecture”.

In this model OCN might serve to inhibit crack increase stretching and dispersing energy in collagen fibers : Please, clarify.

Author Response

Reviewer #3

The myokine Irisin promotes osteogenic differentiation of dental derived MSCs: The authors investigated the effects of Irisin on MSCs from Dental Bud Stem Cells (DBSCs), showing its ability to promote osteoblastic differentiation. In particular, they observed enhanced expression of Osteocalcin (OCN), a late marker of osteoblast differentiation, thus suggesting an improvement of osteoblast mineralizing process in presence of Irisin. The authors propose Irisin as promising molecule participating in osteoblastogenesis process, and conclude that it could be used to improve bone tissue regeneration starting from dental stem cells.

We really appreciate the reviewer’s comments and suggestions and we have tried our best to meet them all.

Major comments:

  1. Too ambitious a conclusion (“The authors propose Irisin as promising molecule participating in osteoblastogenesis process, and conclude that it could be used to improve bone tissue regeneration starting from dental stem cells”), in relation to the simple tests performed in the present work.

Response 1: The reviewer is perfectly right, we have reformulated the sentence in the “Abstract” with: “Considering the in vivo studies about Irisin effects on skeleton, Irisin could improve bone tissue metabolism in MSCs regenerative procedures”.

  1. Bone mineral matrix secretion and production of osteocalcin do not suggest an inductive effect of Irisin in osteoblastogenesis (dependent on the activation of osteoblast precursors to committed osteoblasts) but in improvement of osteoblast differentiation. The authors do not clarify whether there is a positive effect of irisin on increased recruitment of osteoblast precursors, rather than increased/accelerated committed-osteoblast differentiation. The undoubted demonstration is related to the “involvement of Irisin in the osteogenic differentiation process of DBSCs”. Proposed Title: The myokine Irisin promotes osteogenic differentiation of dental bud-derived MSCs.

Response 2: We thank the reviewer for the correct observation, we have changed the title as suggested.

  1. Self-citations: 15 out of the 44 (34%, more than one third of the total). Such a high number of self-citations is fine for a review on a topic historically carried out by the authors, but I don't think it is appropriate for an original article that is quite simple. Moreover, some self-citations are not relevant to the demonstration of irisin effect on osteoblastic differentiation: [23], [19].
  • [19]: … induce the “browning” response stimulating the trans-differentiation of white adipocytes
  • [23]: … effect of Irisin in increasing the polar moment of inertia of bones of mice subjected to Irisin administration.

Response 3: We have removed the citation [19] and the corresponding sentence in the “Introduction” of the manuscript.

We have also deleted the sentence on the “polar moment of inertia” in the discussion section, which referred to the citation [23] (now [22] in the new version of the manuscript). Anyway we cannot completely remove the new citation [22] from the manuscript since, in that paper, Irisin effect on the differentiation of bone marrow stromal cells has been investigated and this aspect is closely connected with the study we are now proposing.

Minor comments:

Introduction and M&M are well described

  1. Results: What is reported in lines 195-199 and 202-204 is more suitable for the discussion paragraph rather than the results paragraph.

Response 4: Following the reviewer’s suggestion we moved those sentences to the discussion section (627-634).

  1. Discussion and Conclusions: Recent studies showed that OCN could influence the bone matrix microarchitecture: Please, explain better. Thus, OCN tightly binds to hydroxyapatite and forms complex structures with collagen and the glycoprotein Osteopontin; OCN could be the link between the organic component of matrix and the mineral fraction: The organic–inorganic interface in bone is not “microarchitecture”. In this model OCN might serve to inhibit crack increase stretching and dispersing energy in collagen fibers : Please, clarify.

Response 5: We thank the reviewer for this request of clarification, we reformulated the period in the Discussion section (lines 673-677): “Recent studies showed that OCN has a crucial role in bone matrix formation since it binds to hydroxyapatite and forms complex with Osteopontin, which is an essential adhesion molecule for both osteoblasts and osteoclasts [39-41]. OCN hence acts as a bridge between calcium crystals and structural proteins, exerting its function at the organic-inorganic interface of bone matrix, and might serve to inhibit crack increase stretching and dispersing energy in collagen fibers [42,43].”

Reviewer 4 Report

The suggestions and comments are in the attached manuscript with revisions.

Author Response

Please find it in the attachment.

Round 2

Reviewer 1 Report

  1. As for Figure 1, it is more accurate to describe that pERK expression was transiently upregulated after Irisin stimulation. Then the authors should rewrite text.

Reviewer 3 Report

The authors answered to the reviewer's requests.
The work has improved.